

# Genome-wide identification and evolution of the SAP gene family in sunflower (*Helianthus annuus* L.) and expression analysis under salt and drought stress

Chun Zhang[1], Xiaohong Zhang[2], Yue Wu[2], Xiang Li[1], Chao Du[2], Na Di[1] and Yang Chen[1]

[1] Hetao College, Bayannur, China
[2] Bayannur Institute of Agriculture and Animal Science, Bayannur, China

## ABSTRACT

Stress-associated proteins (SAPs) are known to play an important role in plant responses to abiotic stresses. This study systematically identified members of the sunflower SAP gene family using sunflower genome data. The genes of the sunflower SAP gene family were analyzed using bioinformatic methods, and gene expression was assessed through fluorescence quantification (qRT-PCR) under salt and drought stress. A comprehensive analysis was also performed on the number, structure, collinearity, and phylogeny of seven Compositae species and eight other plant SAP gene families. The sunflower genome was found to have 27 SAP genes, distributed across 14 chromosomes. The evolutionary analysis revealed that the SAP family genes could be divided into three subgroups. Notably, the annuus variety exhibited amplification of the SAP gene for Group 3. Among the Compositae species, *C. morifolium* demonstrated the highest number of collinearity gene pairs and the closest distance on the phylogenetic tree, suggesting relative conservation in the evolutionary process. An analysis of gene structure revealed that Group 1 exhibited the most complex gene structure, while the majority of HaSAP genes in Group 2 and Group 3 lacked introns. The promoter analysis revealed the presence of cis-acting elements related to ABA, indicating their involvement in stress responses. The expression analysis indicated the potential involvement of 10 genes (*HaSAP1, HaSAP3, HaSAP8, HaSAP10, HaSAP15, HaSAP16, HaSAP21, HaSAP22, HaSAP23*, and *HaSAP26*) in sunflower salt tolerance. The expression of these 10 genes were then examined under salt and drought stress using qRT-PCR, and the tissue-specific expression patterns of these 10 genes were also analyzed. *HaSAP1*, *HaSAP21*, and *HaSAP23* exhibited consistent expression patterns under both salt and drought stress, indicating these genes play a role in both salt tolerance and drought resistance in sunflower. The findings of this study highlight the significant contribution of the SAP gene family to salt tolerance and drought resistance in sunflower.

Corresponding author
Yang Chen, chenyangrz@126.com

## INTRODUCTION

Stress-associated proteins (SAPs) are found in plants and contain a zinc finger domain with an A20 and/or AN1 type zinc finger. These zinc finger domains, namely the N-terminal A20 domain and/or the C-terminal AN1 domain, play crucial roles in the functioning of SAP proteins and their involvement in plant stress responses (*Lee et al., 2000*; *Linnen, Bailey & Weeks, 1993*). The structure of the SAP family is highly conserved across different plant species. The A20 domain typically consists of one or more tandem C2H2 zinc finger structures, while the AN1 sequence is highly conserved (*Gimeno-Gilles et al., 2011*). A previously published bioinformatics analysis of 22 organisms found that some lower organisms only possess the AN1 zinc finger domain and lack the A20 zinc finger domain. This observation suggests that the AN1 zinc finger may have emerged earlier than the A20 zinc finger (*Vij & Tyagi, 2008*). The initial identification and characterization of the AN1 protein were performed in *Xenopus laevis hemisphere 1* (*Vij & Tyagi, 2008*). The first SAP gene to be studied in plants was *OsiSAP1* in rice, which is known to be induced by various stress conditions such as abscisic acid (ABA), drought, low temperature, high salt, and heavy metal stress. Overexpression of *OsiSAP1* in tobacco has been shown to significantly increase the plant's resistance to drought, low temperatures, salinity, and pathogens (*Dansana et al., 2014*). SAP genes have been identified and their functionally has been studied in various plant species, including *Arabidopsis thaliana*, rice, tomato, *Trilicum aestivum* (wheat), *Solanum tuberosum* (potato), and *Hordeum vulgare* (barley; *Vij & Tyagi, 2006*; *Solanke et al., 2009*; *Li et al., 2021*; *Billah et al., 2022*). The evolution of SAP in plants is highly conserved, with a frequent occurrence of intron-free genes. Different species exhibit variations in the zinc finger types of SAP members. For instance, the A20-A20-AN1 zinc finger is found only in rice and eucalyptus, while the A20 type is found in species such as apple, *B. rapa*, and *Vitis vinifera* (grape). The presence or absence of specific zinc finger types in SAP genes within these genomes indicates their essential roles in the survival of these species.

SAP genes have been found to play an important role in plants, especially in plant tolerance to abiotic stresses. Various studies have reported the induction of SAP genes by different stressors and the involvement of SAP genes in the stress responses of different plant species (*Vij & Tyagi, 2006*; *Solanke et al., 2009*; *Li et al., 2021*; *Billah et al., 2022*; *Baidyussen et al., 2020*). In *Arabidopsis thaliana* and *Zea mays*, SAP genes have been found to participate in the plants' response to cold, salt, osmotic, and drought stresses (*Vij & Tyagi, 2006*; *Fu et al., 2022*). Similarly, in rice, all SAP gene groups have been observed to respond to the induction of one or more abiotic stresses, such as drought, cold, or salt stress. Notably, the overexpression of *OsiSAP1* and *OsiSAP8* in both rice and tobacco has been shown to enhance plant resistance against multiple abiotic stresses (*Dansana et al., 2014*; *Kanneganti & Gupta, 2008*). In tomato, an expression analysis of 13 SAP genes showed that all members of the SAP family were upregulated in response to one or more stresses (*Solanke et al., 2009*). Furthermore, overexpression of the *AtSAP5* gene in

*Arabidopsis thaliana* and wheat seedlings has been found to enhance drought resistance (*Hozain et al., 2012*), and overexpression of *AtSAP10* and *AtSAP13* in *Arabidopsis thaliana* has been shown to improve the plant's tolerance to various toxic metals (*Dixit & Dhankher, 2011*; *Dixit et al., 2018*). Similarly, overexpression of SAP16 in both *Arabidopsis thaliana* and soybean plants has been shown to enhance the tolerance of these transgenic plants to drought and salt stress (*Zhang et al., 2019*). Moreover, overexpression of the *M. truncatula SAP1* gene has been found to improve the ability of transgenic tobacco plants to withstand abiotic stresses (*Charrier et al., 2013*). These studies collectively highlight the potential of SAP genes to increase stress tolerance in a wide range of plant species. By understanding the mechanisms and functions of SAP genes, researchers can explore the potential applications of these genes in improving crop resilience to various abiotic stresses.

Sunflower (*Helianthus annuus* L.) plants are known for their drought and salt tolerance and their ability to grow in barren and arid regions (*Park & Burke, 2020*). However, water availability remains a crucial factor affecting sunflower yield. When subjected to drought and salt stress, sunflower plants exhibit visual symptoms such as pale leaf color, slow growth, wilting, and even death (*Shen et al., 2023*; *Song et al., 2024*). Drought and salt stress can also cause physiological changes to occur, including increased proline content, enhanced membrane permeability, and changes in protective enzyme activity. Osmotic stress and ionic toxicity can disrupt ion and water balance within and outside plant cells (*Liu et al., 2020*; *Niu et al., 2022*). Therefore, understanding the mechanisms of drought and salt tolerance and developing stress-resistant sunflower varieties are important research objectives. Genome sequencing has significantly advanced sunflower research, enabling the systematic identification and study of gene families, including the stress-associated protein (SAP) gene family (*Badouin et al., 2017*; *Hübner et al., 2019*). Previous studies have shown the role of SAP genes in enhancing the stress resistance of many plant varieties (*Hozain et al., 2012*; *Dixit & Dhankher, 2011*; *Dixit et al., 2018*; *Charrier et al., 2013*). However, the evolution, function, and taxonomy of the SAP gene family in sunflower plants have not been thoroughly investigated. This study aimed to investigate various aspects of SAP genes in sunflower, including chromosomal distribution, evolutionary relationships, gene structure, promoter cis-acting elements, and expression profiles. This study also analyzed the expression patterns of these genes in different tissues and their responses to salt and drought stress. The expression levels of selected genes were assessed using qRT-PCR in various tissues and under different stress conditions. The findings of this study provide a foundational understanding of how SAP genes are involved in drought resistance and salt tolerance in sunflower plants. The findings of this study provide a better understanding of the genetic factors influencing stress tolerance in sunflower by providing a comprehensive analysis of the SAP gene family. These findings can also serve as a basis for further investigations into the specific functions and regulatory mechanisms of the SAP genes involved in drought resistance and salt tolerance in sunflower plants.

## MATERIALS AND METHODS

### Identification of HaSAP gene family members and analysis of their physicochemical properties

The genomic data of *A. officinalis, O. sativa, Z. mays, A. thaliana, B. rapa, C. lanatus, S. lycopersicum V. vinifera*, and *H. annuus* were downloaded from Ensembl Plants (http://plants.ensembl.org/index.html). *A. lappa, C. endivia*, and *C. intybus* genomic and proteomic data were downloaded from the NCBI database (https://www.ncbi.nlm.nih.gov/). *C. morifolium* genomic and proteomic data were downloaded from the Chrysanthemum Genome Database (http://www.amwayabrc.com/zh-cn/). *C. tinctorius* genomic and proteomic data were downloaded from the NGINX database (http://118.24.202.236:11010/filedown/). The hidden Markov model (HMM) analysis of the SAP gene domain (PF01754 and PF01428) in HMMER 3.2.1 software (http://HMMER.org/) was used to identify SAP protein sequences (*Potter Simon, Aurélien & Eddy Sean, 2018*). The CDD (https://www.ncbi.nlm.nih.gov/Structure/cdd/wrpsb.cgi) database was used to determine whether the candidate sequences had full A20 and/or AN1 functional domains, and the sequences were finalized for subsequent analysis and named according to their chromosomal order (*Shennan, Jiyao & Farideh, 2020*). The physicochemical properties of the HaSAP proteins were predicted using the Swiss Bioinformatics Resource Portal, ExPASy (https://web.ExPASy.org/protparam/; *Julien, Davide & Alessandra, 2018*).

### Phylogenetic analysis of HaSAP

Using the ClustalW default settings in MEGA 7 software, *A. officinalis, O. sativa, Z. mays, A. thaliana, B. rapa, C. lanatus, S. lycopersicum, V. vinifera, H. annuus, A. lappa, C. endivia, C. intybus, C. morifolium, L. sativa var. angustata*, and *C. tinctorius* SAP protein sequences were aligned with multiple sequences. A phylogenetic tree was constructed using the neighbor-joining (NJ) method (the bootstrap method value was set to 1,000 and the remaining parameters were set to default values; *Sudhir, Glen & Koichiro, 2016*) and the EvolView website (https://www.evolgenius.info/evolview/).

### Gene structure and motif analysis of the SAP gene family in Compositae

The motifs of the candidate genes were analyzed using the MEME website (https://meme-suite.org/meme/), with the number of functional domains set to 10. Gene structure information was extracted from the genome database GFF3 file. The xml file obtained from MEME, the nwk file of the evolutionary tree, and the GFF3 file of the gene structure were processed and visualized using TBtools software (*Chen et al., 2023*).

### Analysis of HaSAP cis-acting elements

The 2000 bp sequence upstream of the HaSAP start codon was selected and submitted to the PlantCARE website (http://bioinformatics.psb.ugent.be/webtools/plantcare/html/) to predict the cis-acting elements of the gene promoter region, which were then processed and visualized using TBtools software (*Chen et al., 2023*).

## RNA-seq analysis

The transcriptome sequencing data of sunflower plants under salt stress were downloaded from the NCBI database (PRJNA866668). The raw data were filtered and quality controlled using fastp software (*Chen et al., 2018*). The filtered data were then aligned with the sunflower genome using hisat2 software, and expression quantification was performed using StringTie software (*Pertea et al., 2016*). Gene expression levels were quantified using the fragments per kilobase of transcript sequence per million mapped reads (FPKM) method. To visualize the expression patterns of the genes, an expression heatmap was generated using TBtools software (*Chen et al., 2023*).

## Plant material

The plant material used in this study was the sunflower salt-tolerant inbred line 19S05, which was bred by the Bayannur Institute of Agricultural and Animal Sciences. This particular inbred line was chosen based on its demonstrated tolerance to high salt conditions. To prepare the plant material, the completed 19S05 seeds were washed with water and then disinfected with 3% hydrogen peroxide for 10 min. The plants were subsequently placed in plastic jars filled with disinfected soil. The incubation chamber maintained a temperature of 24 (±2) °C and a humidity level of 60–70%. The plants were exposed to 16 h of light with an intensity of 16,000 lx, followed by 8 h of darkness. An incandescent lamp was used for illumination. After germination, the first pair of true leaves was transplanted into the ordinary hydroponic solution once the leaves were fully expanded. The Hoagland nutrient solution was replaced every 2 days. When the plants reached the stage of having four true leaves, samples of the rhizome leaves were taken. For the salt stress treatment, the nutrient solution was replaced with a 200 mmol/L NaCl solution. Samples were taken at various time points after the treatment: 0, 1, 3, 6, 12, and 24 h. For drought stress, the nutrient solution was replaced with a 15% PEG6000 solution, and samples were taken at the same time points as the salt stress treatment. Each sample was biologically replicated in triplicate, and the samples were flash frozen with liquid nitrogen and stored at −80 °C to preserve the biological material for subsequent analysis.

## qRT–PCR analysis

The FastPure Plant Total RNA Isolation Kit (Novozan Biotechnology Co., Ltd., Nanjing, China) was used to isolate total RNA. After RNA isolation, cDNA synthesis was performed using PrimeScript Reverse Transcriptase (TaKaRa, Dalian, China). For quantitative real-time PCR, primers were designed using Primer Premier 5.0 software. The primer sequences can be found in Table S1. The sunflower *Actin* gene was used as an internal control for normalization purposes. The qRT-PCR was carried out using the ChamQ Universal SYBR qPCR Master Mix (Novozan Biotechnology Co., Ltd., Nanjing, China). The qRT-PCR protocol involved an initial denaturation step at 95 °C for 30 s, followed by denaturation at 95 °C for 10 s, annealing at 60 °C for 30 s, and extension at 72 °C for 20 s. This cycle was repeated for a total of 40 cycles. All PCRs were performed in triplicate to ensure the reliability of the results. The quantitative data obtained from qRT-PCR were analyzed using the $2^{-\Delta\Delta CT}$ method (*O'Connell et al., 2017*). Excel 2010 was also used for
statistical analysis of the qRT-PCR data. The R language (version 4.2.3) ggplot2 software package was used for data visualization. Additionally, the ggpubr software package was used to perform a t-test on the data, specifically comparing the samples before 0 h (CK) and after stress, and the significance level was calculated to determine if there were statistically significant differences in gene expression between the control and stress conditions.

# RESULTS

## Identification of the SAP gene family in sunflowers

To study changes in the copy number of the SAP gene family during the evolution of Compositae plants, the SAP genes were comprehensively searched from the genomes of seven Compositae species (*H. annuus, A. lappa, C. endivia, C. intybus, C. morifolium, L. sativa var. angustata*, and *C. tinctorius*) *via* hmmsearch. The results of the search were verified in the NCBI-CDD database. A total of 27, 17, 19, 14, 18, 15, and 11 SAP sequences were identified, respectively. The 27 sunflower sequences on the chromosome were named HaSAP01–HaSAP27 (Table 1). The open reading frame (OFR) length of the HaSAP family gene ranged from 402 to 873 bp. The encoded protein contained between 133 and 290 amino acid residues, with a relative molecular weight ranging from 14.89 to 31.74 kDa. The theoretical isoelectric point of these proteins ranged from 7.85 to 9.71. In terms of genomic distribution, 27 HaSAP genes were found on 14 chromosomes of sunflower (chr1, chr3, chr4, chr6, chr7, chr8, chr9, chr10, chr11, chr12, chr13, chr14, chr16, and chr17). Among those chromosomes, chr11 contained the highest number of HaSAP genes, with six genes. Chr14 contained four HaSAP genes, while chr1, chr7, chr8, chr10, and chr13 each contained two HaSAP genes. The remaining chromosomes (chr3, chr4, chr6, chr9, chr12, chr16, and chr17) each contained one HaSAP gene.

## Phytogenetic analysis of the SAP gene family in sunflower

To better understand the evolutionary relationship of sunflower SAP family genes, a phylogenetic tree was constructed based on 238 SAP protein sequences from seven Compositae species (*H. annuus, A. lappa, C. endivia, C. intybus, C. morifolium, L. sativa var. Angustata*, and *C. tinctorius*), as well as eight other plant species (*A. officinalis, O. sativa, Z. mays, A. thaliana, B. rapa, C. lanatus, S. lycopersicum, V. vinifera*). The evolutionary tree was divided into three distinct groups (Fig. 1A). Group 1 consisted of 72 SAP genes, with *C. endivia* and *B. rapa* having the highest number of SAP genes (8 and 7, respectively). *A. thaliana, C. morifolium*, and *H. annuus* each had six SAP genes, while *Z. mays* only had two SAP genes (Fig. 1B). Group 2 contained a total of 68 SAP genes, with *B. rapa* having the highest number (15), while *A. officinalis* did not have any SAP genes. *A. thaliana, A. lappa*, and *H. annuus* each had six SAP genes. Group 3 consisted of 98 SAP genes, with *H. annuus* having the highest number (15), and *A. thalianas* having only two SAP genes. *O. sativa* and *C. morifolium* each contained nine SAP genes. Notably, *B. rapa*, despite having the highest number of SAP genes overall (29), was mainly distributed in Group 2. The phylogenetic tree results indicated that the SAP genes of the seven Compositae plants were more closely related within each branch. Additionally, the number
**Table 1 Sunflower SAP gene family member information.**

| Gene name | Gene id | Open reading frame/bp | Protein length/aa | Relative molecular weight (r)/kDa | Theoretical isoelectric point (pI) | Chromosome location |
|---|---|---|---|---|---|---|
| HaSAP1 | HanXRQr2_Chr01g0012971 | 837 | 278 | 30.40 | 8.21 | chr1:59710103-59712466 |
| HaSAP2 | HanXRQr2_Chr01g0026611 | 477 | 158 | 17.58 | 8.72 | chr1:105451161-105451965 |
| HaSAP3 | HanXRQr2_Chr03g0131611 | 510 | 169 | 17.92 | 8.42 | chr3:166637451-166640007 |
| HaSAP4 | HanXRQr2_Chr04g0162551 | 504 | 167 | 17.66 | 8.27 | chr4:90388855-90391859 |
| HaSAP5 | HanXRQr2_Chr06g0239681 | 462 | 153 | 16.70 | 8.72 | chr6:2309799-2310425 |
| HaSAP6 | HanXRQr2_Chr07g0300041 | 495 | 164 | 17.89 | 8.42 | chr7:113724033-113724878 |
| HaSAP7 | HanXRQr2_Chr07g0304921 | 468 | 155 | 17.16 | 9.18 | chr7:126846796-126847575 |
| HaSAP8 | HanXRQr2_Chr08g0328831 | 507 | 168 | 18.29 | 8.38 | chr8:22597092-22597901 |
| HaSAP9 | HanXRQr2_Chr08g0339381 | 411 | 136 | 15.14 | 9.11 | chr8:57517545-57517955 |
| HaSAP10 | HanXRQr2_Chr09g0402491 | 666 | 221 | 24.41 | 9.11 | chr9:160504365-160505517 |
| HaSAP11 | HanXRQr2_Chr10g0433481 | 633 | 210 | 22.51 | 8.05 | chr10:61719246-61723474 |
| HaSAP12 | HanXRQr2_Chr10g0433941 | 873 | 290 | 31.47 | 8.9 | chr10:65483981-65486466 |
| HaSAP13 | HanXRQr2_Chr11g0474691 | 462 | 153 | 16.73 | 8.72 | chr11:16460312-16460773 |
| HaSAP14 | HanXRQr2_Chr11g0476111 | 438 | 145 | 16.33 | 8.77 | chr11:20557190-20557671 |
| HaSAP15 | HanXRQr2_Chr11g0498351 | 477 | 158 | 17.46 | 9.71 | chr11:124647832-124649071 |
| HaSAP16 | HanXRQr2_Chr11g0498361 | 513 | 170 | 18.30 | 8.21 | chr11:124671225-124681661 |
| HaSAP17 | HanXRQr2_Chr11g0498391 | 513 | 170 | 18.30 | 8.21 | chr11:124746910-124749236 |
| HaSAP18 | HanXRQr2_Chr11g0498401 | 510 | 169 | 18.06 | 7.85 | chr11:124749238-124751974 |
| HaSAP19 | HanXRQr2_Chr12g0528291 | 576 | 191 | 21.59 | 8.66 | chr12:13265031-13265736 |
| HaSAP20 | HanXRQr2_Chr13g0580381 | 426 | 141 | 15.17 | 8.77 | chr13:67896332-67896757 |
| HaSAP21 | HanXRQr2_Chr13g0589391 | 705 | 234 | 26.25 | 8.93 | chr13:92129218-92130772 |
| HaSAP22 | HanXRQr2_Chr14g0657111 | 459 | 152 | 17.04 | 8.69 | chr14:149655111-149655813 |
| HaSAP23 | HanXRQr2_Chr14g0660821 | 480 | 159 | 17.28 | 8.36 | chr14:157564421-157564900 |
| HaSAP24 | HanXRQr2_Chr14g0669051 | 411 | 136 | 15.28 | 8.64 | chr14:170604046-170604456 |
| HaSAP25 | HanXRQr2_Chr14g0669061 | 402 | 133 | 14.89 | 8.94 | chr14:170611582-170611983 |
| HaSAP26 | HanXRQr2_Chr16g0759421 | 417 | 138 | 15.67 | 9.15 | chr16:153746976-153747563 |
| HaSAP27 | HanXRQr2_Chr17g0817861 | 477 | 158 | 17.67 | 8.72 | chr17:139420663-139421368 |

of SAP genes in the other seven Compositae plants showed similarity, except for *H. annuus* in Group 3, which suggests that the SAP genes of *H. annuus* in Group 3 had undergone amplification.

The sunflower SAP gene was used as the core gene to explore collinear relationships of SAP genes between *H. annuus* and *A. officinalis, O. sativa, Z. mays, A. thaliana, B. rapa, C. lanatus, S. lycopersicum*, and *V. vinifera* (Fig. 2). *H. annuus* produced three pairs of collinearity genes with two SAP genes from *A. officinalis*, two pairs of collinearity genes with two SAP genes from *A. thaliana*, eight pairs of collinearity genes with six SAP genes from *B. rapa*, one pair of collinearity genes with one SAP gene from *C. lanatus*, seven pairs of collinearity genes with four SAP genes from *O. sativa*, six pairs of collinearity genes with two SAP genes from *S. lycopersicum*, seven pairs of collinearity genes with three SAP genes

a

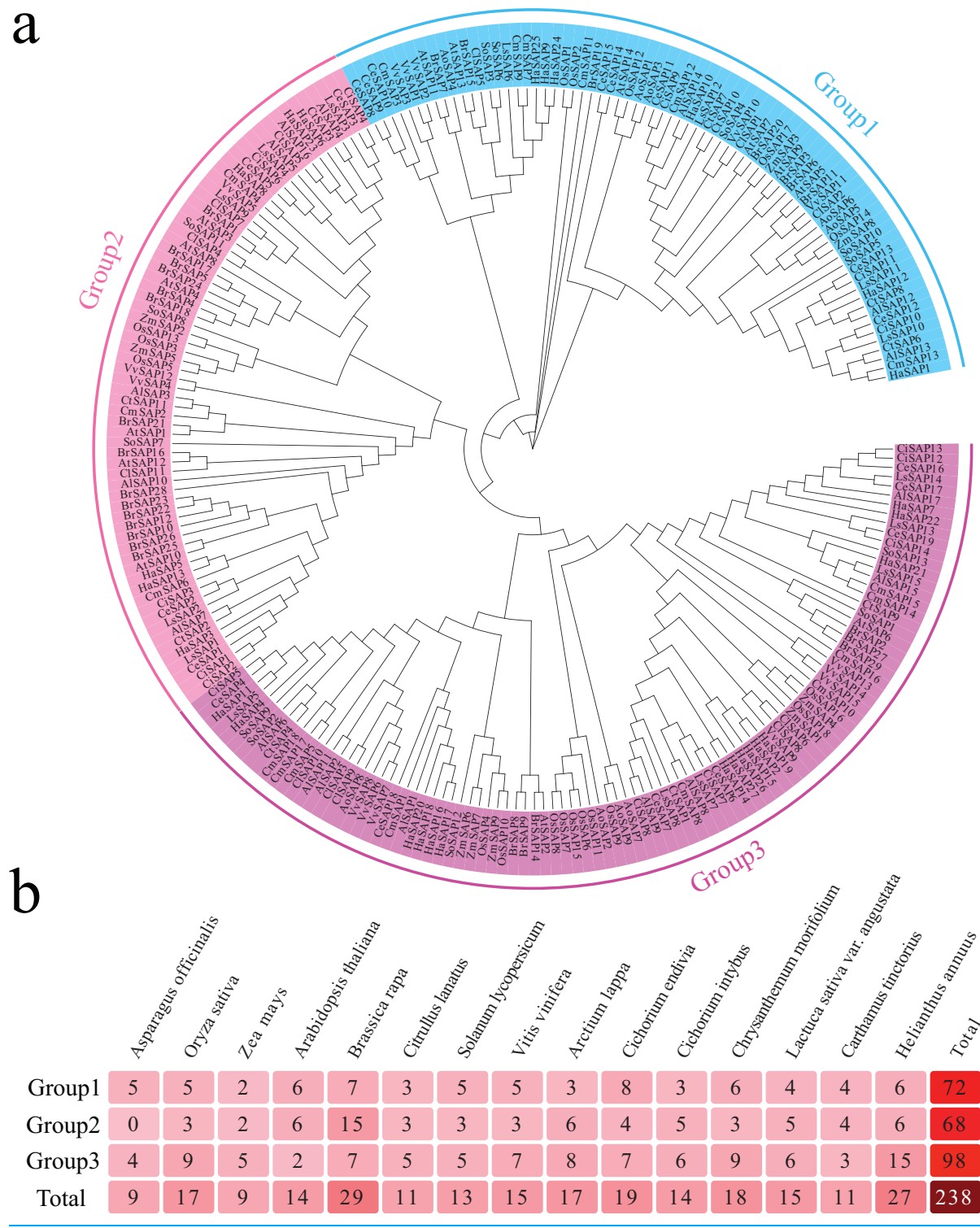

| | *Asparagus officinalis* | *Oryza sativa* | *Zea mays* | *Arabidopsis thaliana* | *Brassica rapa* | *Citrullus lanatus* | *Solanum lycopersicum* | *Vitis vinifera* | *Arctium lappa* | *Cichorium endivia* | *Cichorium intybus* | *Chrysanthemum morifolium* | *Lactuca sativa* var. *angustata* | *Carthamus tinctorius* | *Helianthus annuus* | Total |
|---|---|---|---|---|---|---|---|---|---|---|---|---|---|---|---|---|
| Group1 | 5 | 5 | 2 | 6 | 7 | 3 | 5 | 5 | 3 | 8 | 3 | 6 | 4 | 4 | 6 | 72 |
| Group2 | 0 | 3 | 2 | 6 | 15 | 3 | 3 | 3 | 6 | 4 | 5 | 3 | 5 | 4 | 6 | 68 |
| Group3 | 4 | 9 | 5 | 2 | 7 | 5 | 5 | 7 | 8 | 7 | 6 | 9 | 6 | 3 | 15 | 98 |
| Total | 9 | 17 | 9 | 14 | 29 | 11 | 13 | 15 | 17 | 19 | 14 | 18 | 15 | 11 | 27 | 238 |

**Figure 1  Evolutionary tree analysis.** (A) Phylogenetic tree based on 238 SAP protein sequences from seven Compositae species (*H. annuus, A. lappa, C. endivia, C. intybus, C. morifolium, L. sativa* var. *Angustata,* and *C. tinctorius*) and eight other plant species (*A. offcinalis, O. sativa, Z. mays, A. thaliana, B. rapa, C. lanatus, S. lycopersicum,* and *V. vinifera*), with different colors indicating different groups. (B) Number of genes in each group of the phylogenetic tree of the seven Compositae species (*H. annuus, A. lappa, C. endivia, C. intybus, C. morifolium, L. sativa* var. *Angustata,* and *C. tinctorius*) and eight other plant species (*A. offcinalis, O. sativa, Z. mays, A. thaliana, B. rapa, C. lanatus, S. lycopersicum,* and *V. vinifera*).

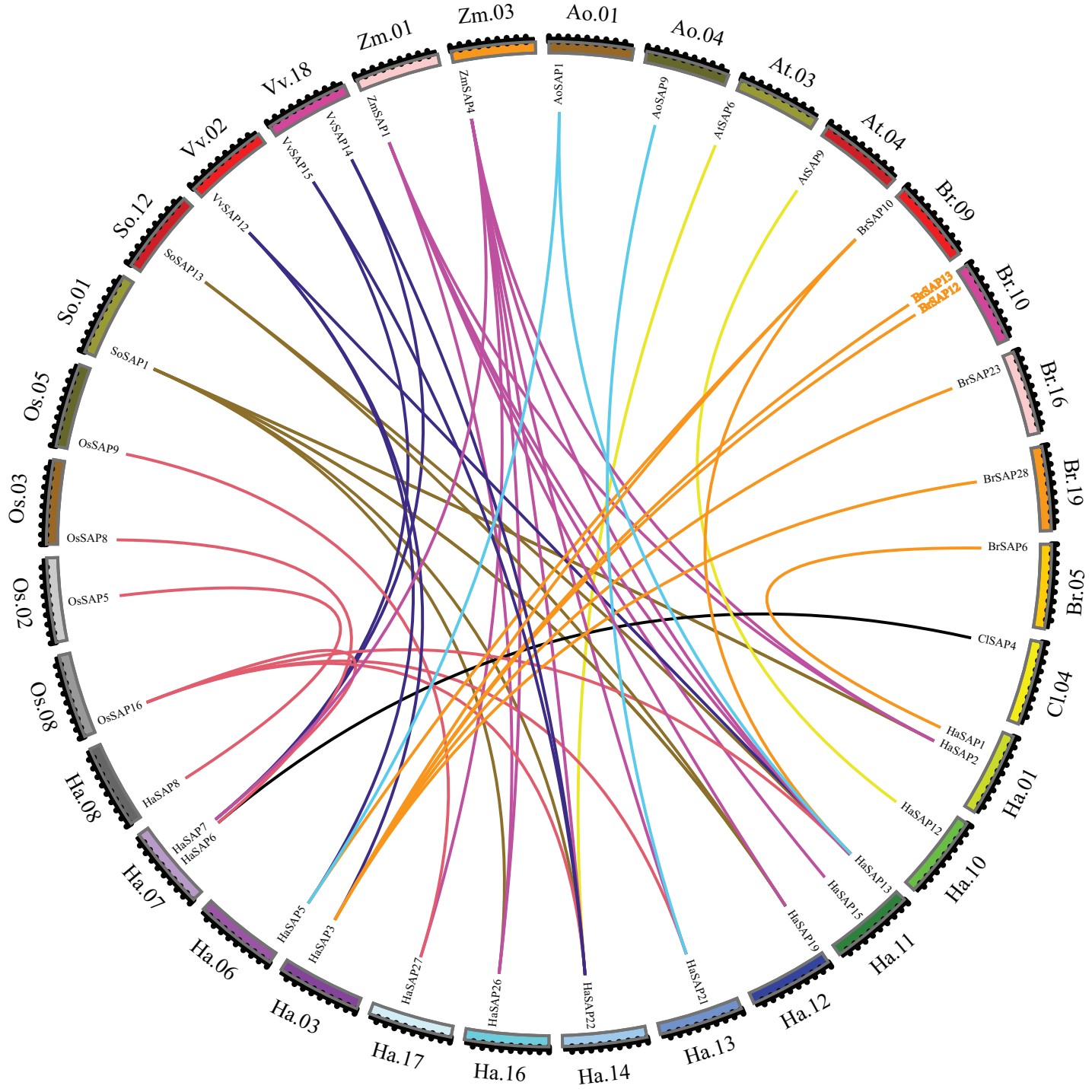

**Figure 2 Collinearity analysis.** Collinear relationships of SAP genes of *H. annuus* and *A. offcinalis, O. sativa, Z. mays, A. thaliana, B. rapa, C. lanatus, S. lycopersicum,* and *V. vinifera*. The different colors represent the collinear relationships between different plant species.

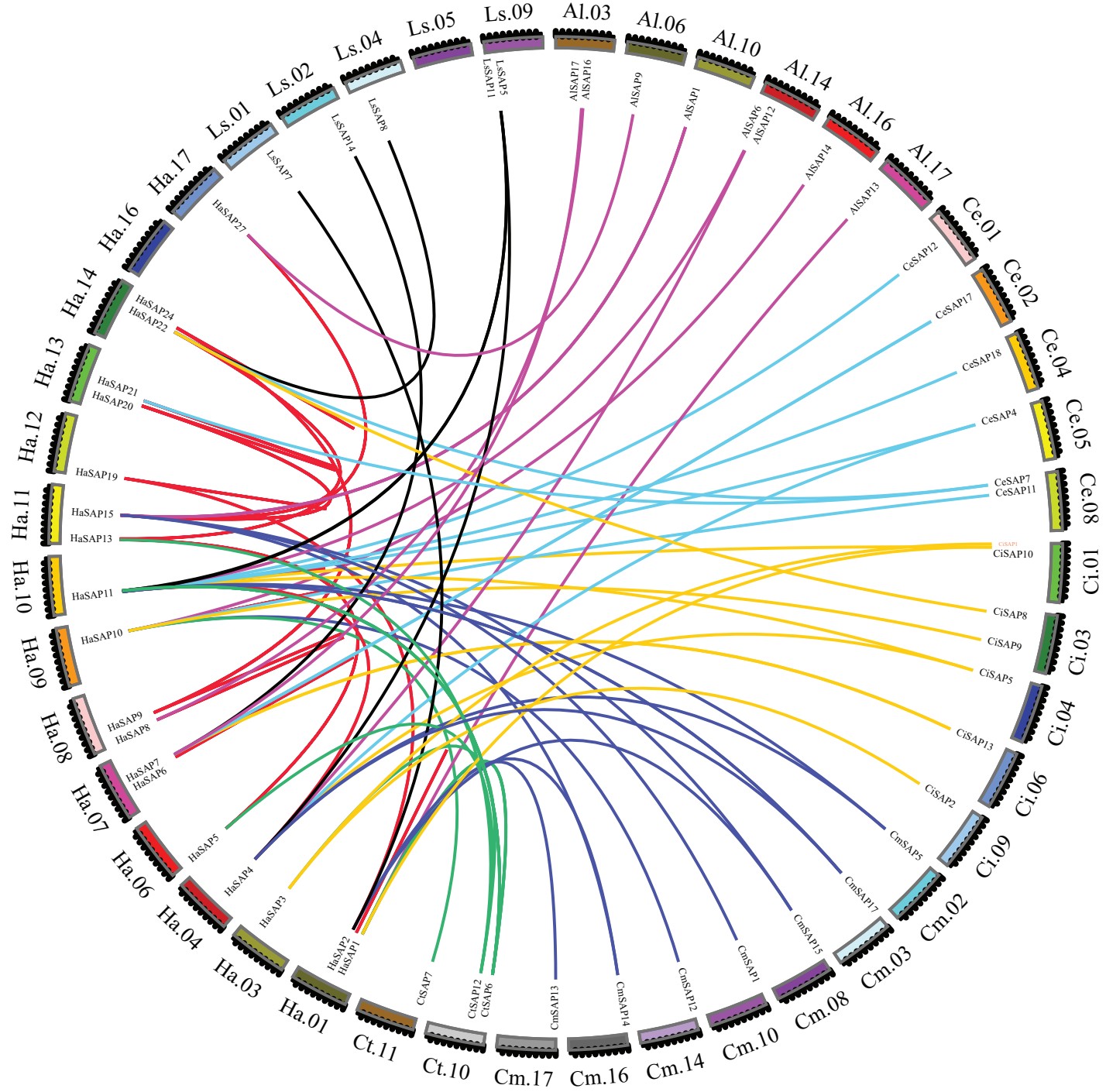

**Figure 3 Collinearity analysis.** Collinearity of SAP genes in seven Compositae species (*H. annuus, A. lappa, C. endivia, C. intybus, C. morifolium, L. sativa var. angustata,* and *C. tinctorius*). The different colors represent the collinear relationships between different plant species.

from *V. vinifera*, and 12 pairs of collinearity genes with two SAP genes from *Z. mays*. Although the most collinearity gene pairs were found between *H. annuus* SAP and *Z. mays*, only two ZmSAP genes (*ZmSAP1* and *ZmSAP4*) were formed.

The collinear relationships of SAP genes were also analyzed for seven Compositae plants (*H. annuus, A. lappa, C. endivia, C. intybus, C. morifolium, L. sativa var. Angustata*, and *C. tinctorius*; Fig. 3). The *H. annuus* genome contains 16 collinearity SAP gene pairs. *H. annuus* produced 10 collinearity pairs with eight SAP genes from *A. lappa*, nine collinearity pairs with six SAP genes from *C. endivia*, 10 collinearity pairs with six SAP genes from *C. intybus*, 12 collinearity pairs with seven SAP genes from *C. morifolium*, six collinearity pairs with three SAP genes from *C. tinctorius*, and eight collinearity pairs with five SAP genes from *L. sativa*. *H. annuus* contained the most collinearity gene pairs with *C. morifolium*, and *H. annuus* and *C. morifolium* had the closest distance in the phylogenetic tree. The collinearity of SAP gene pairs between sunflower and other Compositae plants was not much different, and the number of branches in the phylogenetic tree was relatively similar, indicating that the SAP gene family was conserved in the evolution of Compositae plants.

## Evolution, gene structure, and motif analysis of the Compositae species SAP gene family

A phylogenetic tree was created and gene structure and motif analyses were performed based on the full-length, CDS, and protein sequences of SAP genes in seven Compositae plants to further understand the composition and structure of sunflower SAP genes (Fig. 4). The phylogenetic tree revealed that the SAP genes from the seven Compositae plants could be divided into three subgroups. This subdivision in subgroups is consistent with similar research conducted on other plant species, indicating a certain level of evolutionary conservation within the SAP gene family. A motif is a structural component of a protein molecule with a specific spatial conformation and a specific function and is a subunit of the domain. Members of the same subgroup exhibited similar motifs. Groups 2 and 3 displayed the simplest gene structure and the least number of motifs. Except for CmSAP3, the SAP genes in Group 1 contained at least two exons and four motifs (motif 4, motif 5, motif 6, and motif 7). In Group 2, with a few exceptions (*CiSAP1, CeSAP1, LsSAP1, HaSAP3, HaSAP6, HaSAP9*, and *HaSAP24*), the SAP genes contained one exon and four motifs (motif 1, motif 3, motif 4, and motif 9). In Group 3, excluding a few genes that contained two exons (*HaSAP7, HaSAP11, HaSAP17, CmSAP5, CmSAP11, CmSAP15, CmSAP18, CeSAP6, CeSAP17, CiSAP7, LsSAP8*, and *AlSAP15*), the remaining SAP genes had one exon and seven motifs (motif 1, motif 2, motif 3, motif 5, motif 8, motif 9, and motif 10). These findings suggest that the three subgroups of SAP genes may have undergone genetic structure changes during the evolutionary process, potentially resulting in different biological functions.

## Analysis of promoter cis-acting elements of the SAP gene family in sunflower

The analysis of the 2000 bp upstream promoter region of the HaSAP gene provided insights into its potential functions. The cis-acting element analysis revealed an uneven distribution of cis-acting elements associated with stress and hormonal responses. Four cis-acting elements related to stress responses were identified; among them, the most

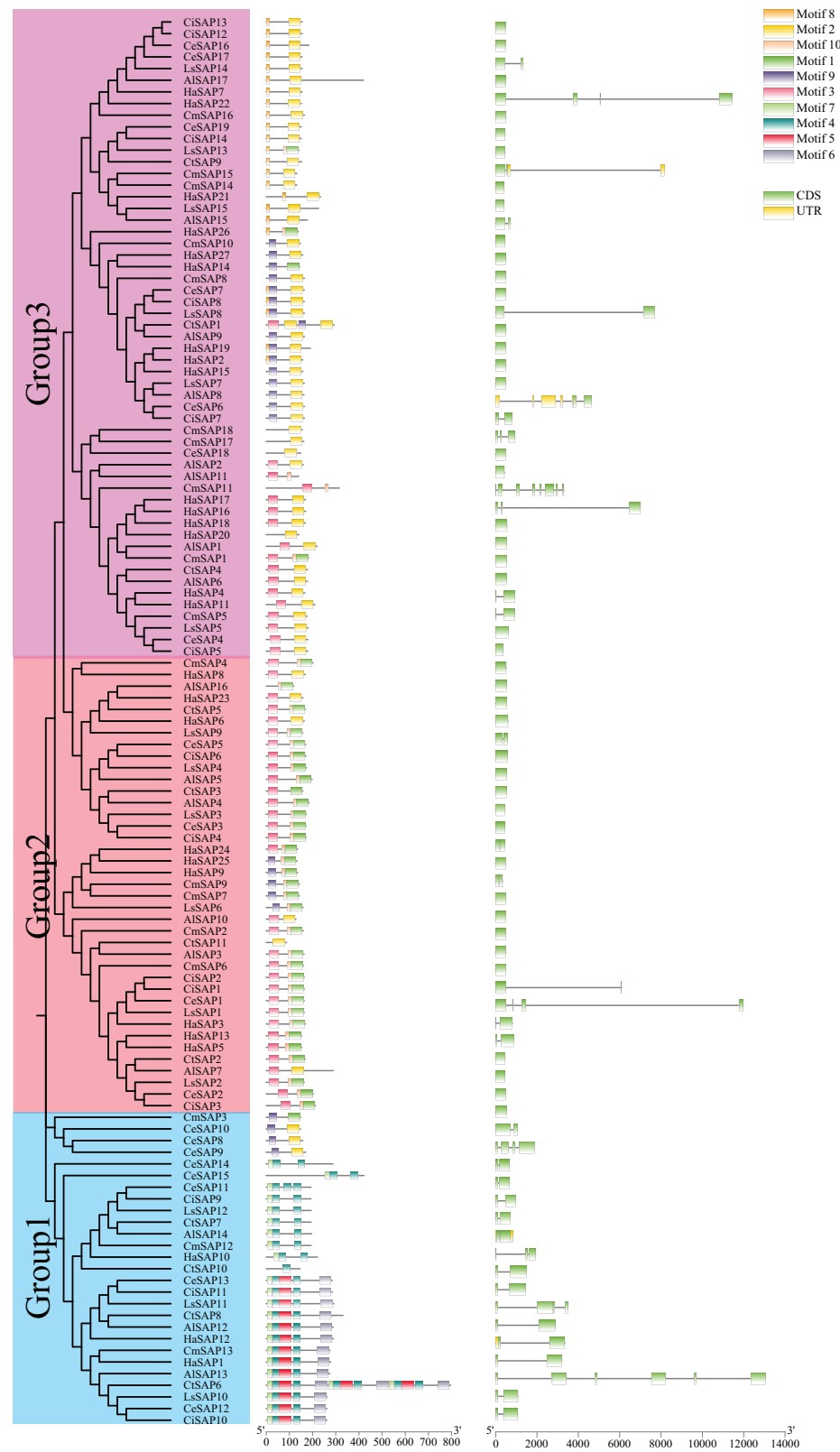

**Figure 4 Phylogenetic tree, gene structure, and motif analysis of the SAP family in seven Compositae plants (*H. annuus, A. lappa, C. endivia, C. intybus, C. morifolium, L. sativa var. angustata*, and *C. tinctorius*).**

abundant element was the anaerobic response element (ARE), which plays a crucial role in anaerobic induction. The majority of HaSAP genes (25 out of 27) contained at least one ARE. Additionally, *HaSAP18* had eight low-temperature responsiveness (LTR) elements, suggesting its potential involvement in low-temperature tolerance. The promoter region of the HaSAP gene contained nine cis-acting elements associated with five hormones: auxin (TGA-element and AuxRR-core), abscisic acid (ABRE), MeJA (TGACG-motif and CGTCA-motif), gibberellin (P-box, GARE-motif and TATC-box), and salicylic acid (TCA-element; Fig. 5). These cis-acting elements suggest the HaSAP gene is involved in various hormone signaling pathways. Notably, the largest number of cis-acting elements was associated with abscisic acid responses, with a total of 64 elements identified, suggesting that the HaSAP gene family may primarily function through synergistic effects on ABA biosynthesis or signal transduction genes.

## Analysis of the expression patterns of the SAP gene family in sunflower plants under salt stress

The expression patterns of HaSAP genes were investigated by analyzing RNA-seq data under salt stress conditions (Fig. 6). Out of all the HaSAP genes analyzed, a subset of 10 genes (*HaSAP1, HaSAP3, HaSAP8, HaSAP10, HaSAP15, HaSAP16, HaSAP21, HaSAP22, HaSAP23*, and *HaSAP26*) exhibited significant changes in expression in response to salt stress. These 10 genes showed specific induction upon exposure to high salt levels, while the remaining HaSAP genes either displayed minor changes in expression or remained relatively stable under salt stress conditions. This suggests that these 10 genes may be specifically involved in the plant's response to salt stress. The expression levels of the five genes with the most significant changes (*HaSAP3, HaSAP10, HaSAP15, HaSAP16*, and *HaSAP26*) reached their minimums at 3 h. However, since RNA-seq data were only measured for a duration of 12 h, it is possible that the expression levels of these genes continued to change beyond the 12-h time point. These results showed that approximately one-third of the HaSAP family genes demonstrated significant changes in expression levels under salt stress conditions, indicating that the HaSAP family genes may play a role in the plant's response to salt stress.

## qRT-PCR analysis of candidate gene expression under drought and salt stress and tissue-specific expression

The qRT-PCR analysis of the selected genes (*HaSAP1, HaSAP3, HaSAP8, HaSAP10, HaSAP15, HaSAP16, HaSAP21, HaSAP22, HaSAP23*, and *HaSAP26*) under salt stress conditions validated the findings of the expression analysis. The transcript levels of the selected genes were measured at different time points after stress treatment. Compared to the expression levels at 0 h under salt stress, significant changes in transcript levels were observed for all ten genes (Fig. 7). Four genes (*HaSAP1, HaSAP3, HaSAP16*, and *HaSAP26*) showed a significant decrease in expression after salt stress, reaching their minimum levels at 6 or 12 h, while five genes (*HaSAP8, HaSAP10, HaSAP21, HaSAP22*, and *HaSAP23*) exhibited a significant increase in expression after salt stress, reaching their

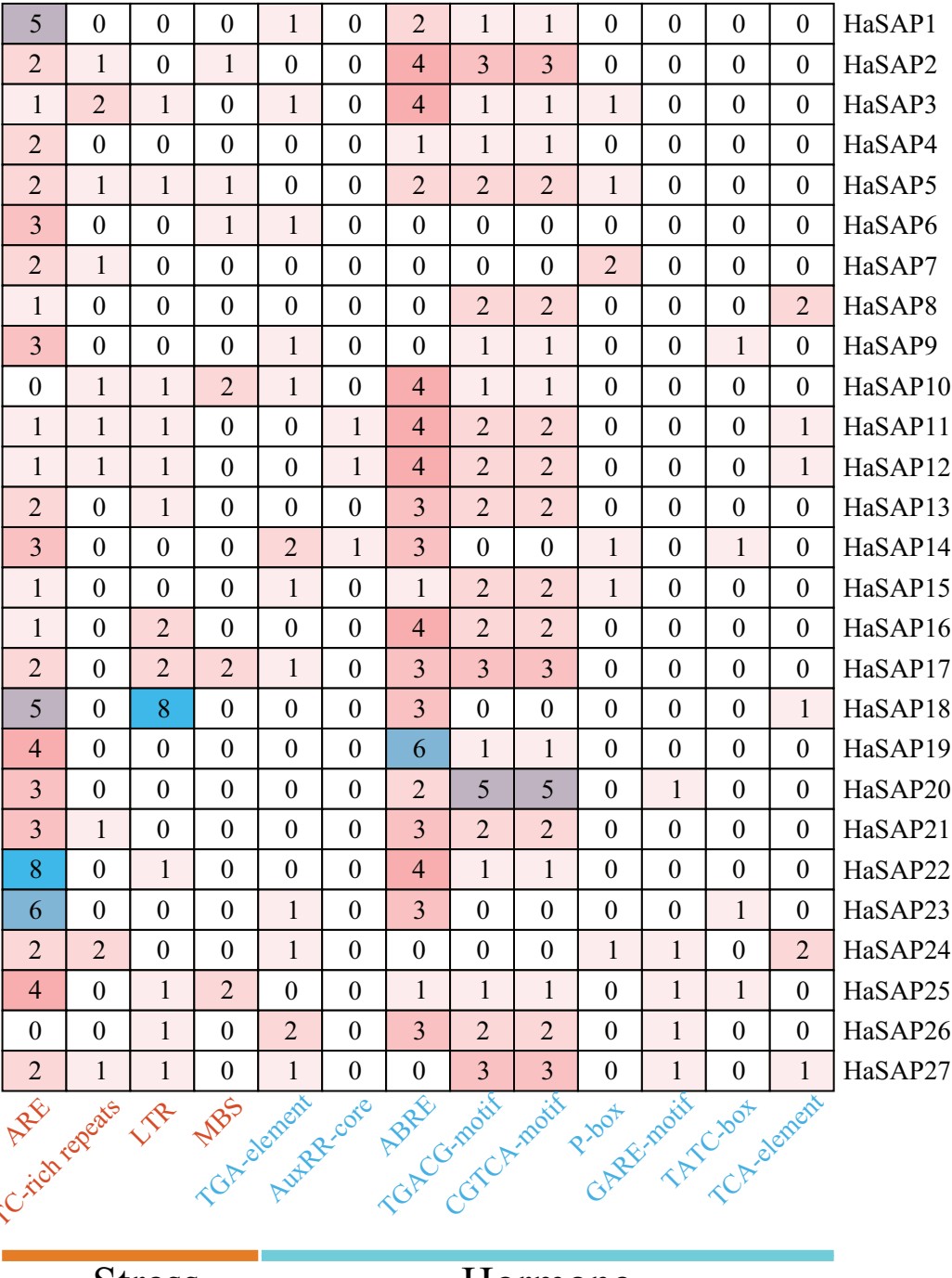

**Figure 5 The number of cis-acting elements in the promoter region of the HaSAP gene, including the type and number of cis-acting elements involved in stress response and hormone response.**

maximum levels at 1, 6, or 12 h. The expression of *HaSAP15* under salt stress was particularly notable as it showed a significant increase after 1 h of salt stress, followed by a significant decrease after 3 h, reaching its lowest expression level at 6 h.

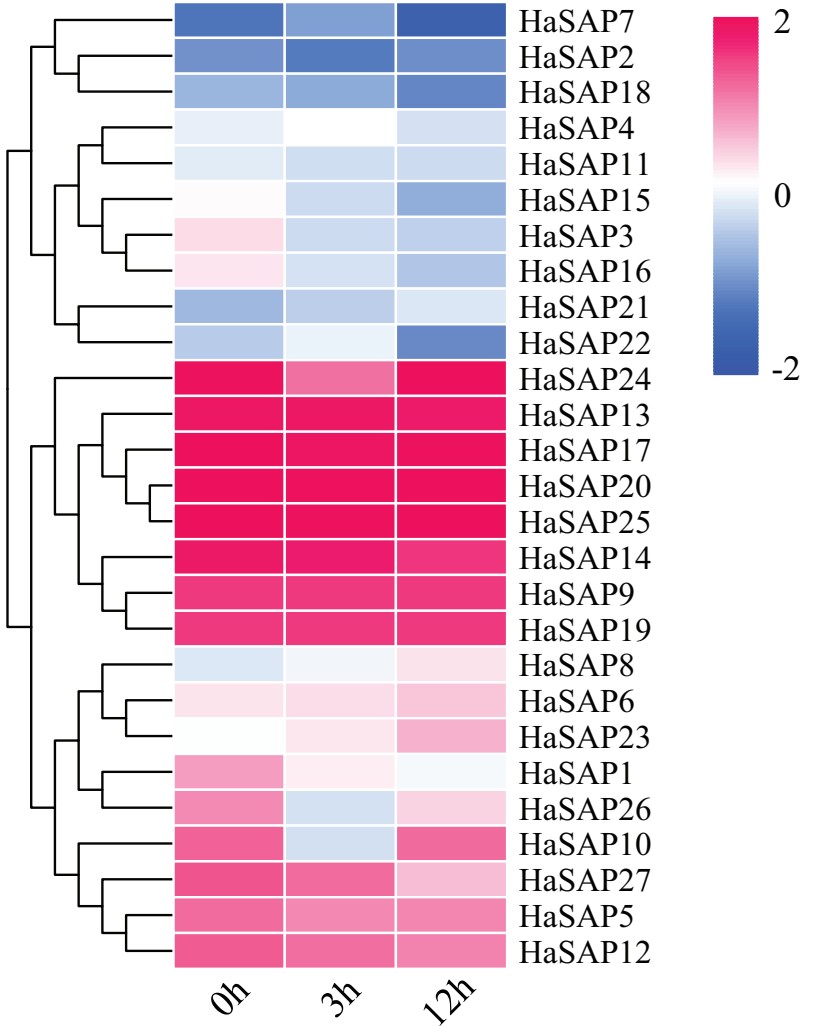

**Figure 6** RNA-seq expression analysis of the HaSAP genes under salt stress conditions.

The expression patterns of the 10 genes (*HaSAP1, HaSAP3, HaSAP8, HaSAP10, HaSAP15, HaSAP16, HaSAP21, HaSAP22, HaSAP23,* and *HaSAP26*) were further investigated under drought stress conditions using qRT-PCR analysis (Fig. 8). Under drought stress, eight genes (*HaSAP1, HaSAP3, HaSAP8, HaSAP15, HaSAP16, HaSAP21, HaSAP23,* and *HaSAP26*) exhibited significant increases in transcript levels compared to their expression levels at 0 h, indicating that these eight genes were strongly induced at the transcriptional level in response to drought stress. Under salt stress, the expression levels of two genes (*HaSAP1* and *HaSAP8*) significantly decreased, reaching their lowest levels at 6 h, while the expression levels of six genes (*HaSAP3, HaSAP15, HaSAP16, HaSAP21, HaSAP23,* and *HaSAP26*) increased significantly after salt stress. *HaSAP1, HaSAP21,* and *HaSAP23* all displayed consistent expression patterns under both salt and drought stress conditions. These findings suggest that these three genes may play crucial roles in the mechanisms of salt tolerance and drought resistance in sunflowers.

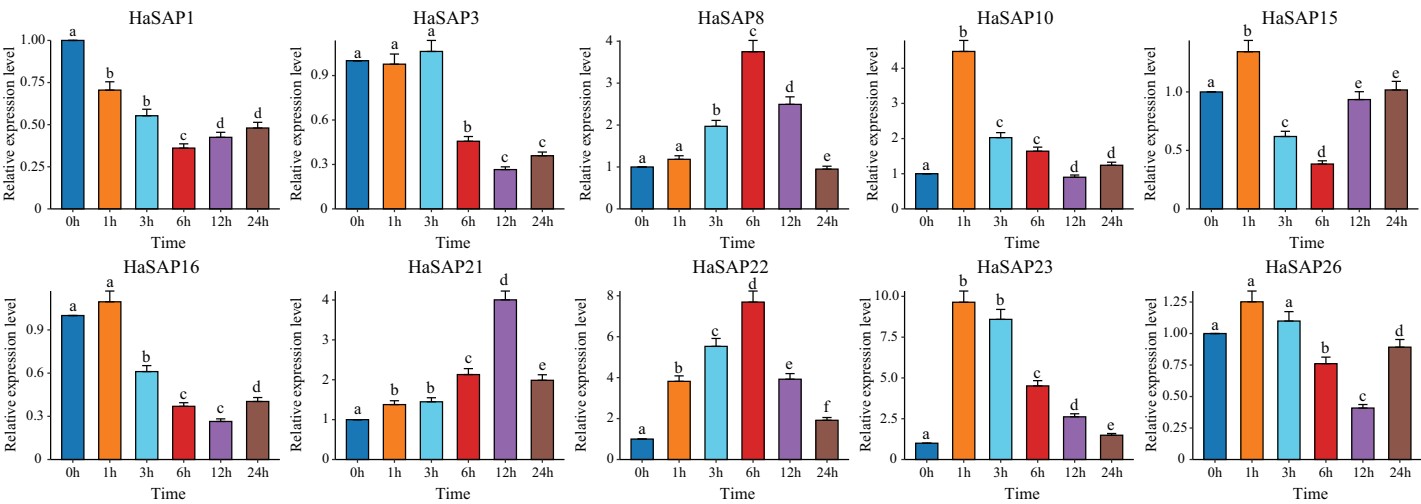

**Figure 7 Analysis of the expression pattern of the HaSAP gene under salt stress.** The error bars represent the mean ± SEs of the three replicates. Different lowercase letters indicate significant differences ($P < 0.05$).

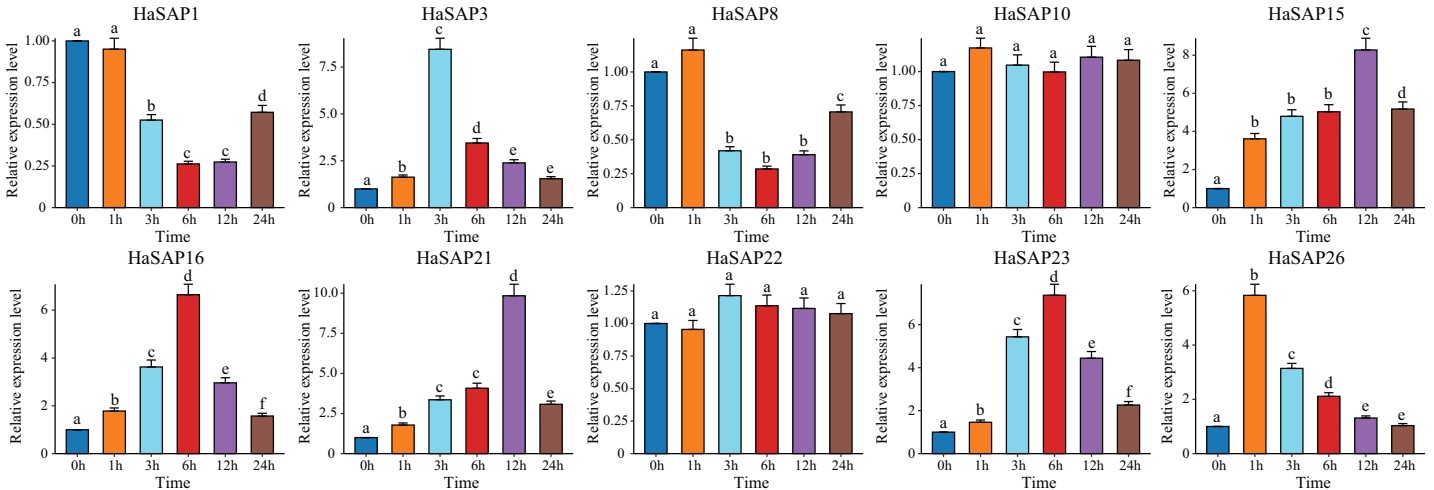

**Figure 8 The expression pattern of the HaSAP gene was analyzed under drought stress.** The error bars represent the mean ± SEs of the three replicates. Different lowercase letters indicate significant differences ($P < 0.05$).

Tissue-specific expression patterns of the 10 genes (*HaSAP1, HaSAP3, HaSAP8, HaSAP10, HaSAP15, HaSAP16, HaSAP21, HaSAP22, HaSAP23,* and *HaSAP26*) were also analyzed in the roots, stems, and leaves (Fig. 9). Six genes (*HaSAP1, HaSAP3, HaSAP15, HaSAP16, HaSAP22,* and *HaSAP26*) displayed significantly higher expression levels in the roots compared to levels in the stems and leaves, indicating that these genes are likely involved in root-specific functions, including some functions related to the plant's response to salt stress. Conversely, the *HaSAP8, HaSAP10,* and *HaSAP23* genes exhibited significantly higher expression levels in the leaves compared to levels in the roots and stems, indicating that these genes may play specific roles in leaf-related processes, including some processes associated with the plant's response to drought stress. Notably, the *HaSAP21* gene displayed the lowest expression in stems, and had similar expression

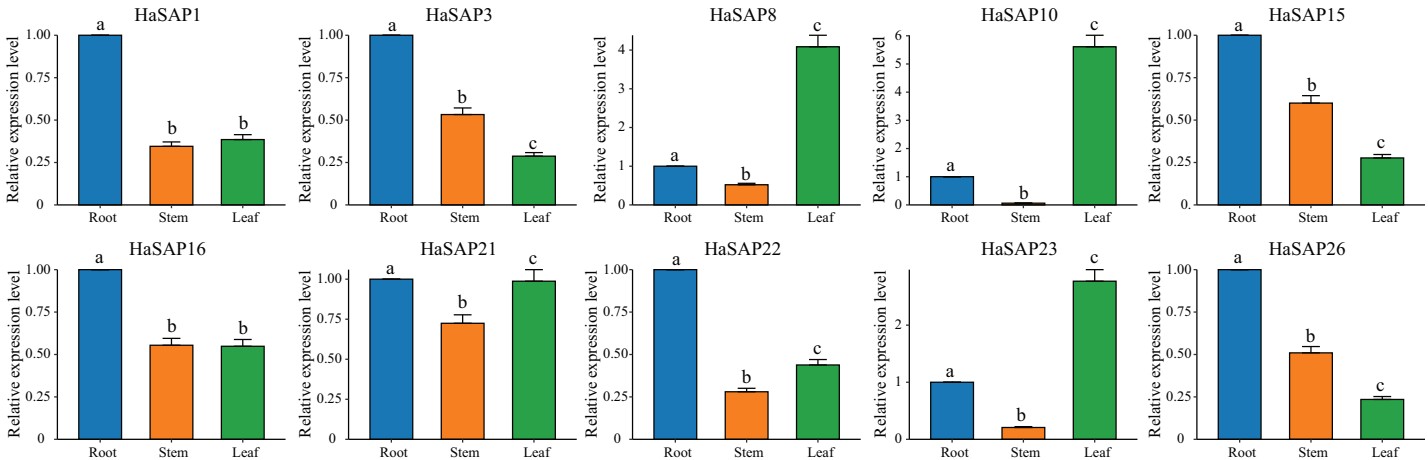

**Figure 9 Tissue-specific expression analysis of the HaSAP gene.** The error bars represent the mean ± SEs of the three replicates. Different lowercase letters indicate significant differences (*P* < 0.05).

levels in the roots and leaves. This suggests that *HaSAP21* may have a more uniform expression pattern across different tissues than the other genes. The significant changes observed in the expression of HaSAP genes in plant leaves and roots after stress treatment likely contribute to improved drought and salt tolerance in sunflower plants. These findings highlight the importance of the HaSAP gene family in the stress tolerance traits of sunflower plants. These findings also provide a foundation for validating the functional roles of these genes and investigating the underlying molecular mechanisms involved in plant stress resistance.

## DISCUSSION

SAPs, which are characterized by zinc finger proteins with an A20/ANI domain, have been identified in various species, including rice, *A. thaliana*, and maize. These genes have been found to play a role in plant responses to both biotic and abiotic stresses. Research on SAP genes and their functions in stress resistance has gained considerable attention in the scientific community (*Dansana et al., 2014*; *Vij & Tyagi, 2006*; *Fu et al., 2022*). Understanding the SAP gene family across different plant species contributes to the knowledge of plant stress adaptation strategies and provides opportunities for improving crop resilience to environmental stresses (*Dansana et al., 2014*; *Vij & Tyagi, 2006*; *Solanke et al., 2009*; *Li et al., 2021*; *Billah et al., 2022*). In this study, a total of 27 SAP genes were identified in sunflower, which is a higher number of SAP genes than has been found in other species such as *A. officinalis, O. sativa, Z. mays, A. thaliana, C. lanatus, S. lycopersicum, V. vinifera*, and *H. annuus*. A conserved domain analysis revealed that all the sunflower SAP proteins contained AN1 domains, with 74% of the genes being A20-AN1 zinc finger proteins, similar to SAP family members in other species (Fig. S1). The absence of SAP genes with only the A20 domain in *Arabidopsis thaliana* and grape and the presence of only one gene (*OsSAP18*) with the A20 domain in rice suggests that the AN1 domain may be older than the A20 domain in terms of evolutionary origins (*Vij & Tyagi, 2006*; *Sun, Xia & Guan, 2022*). This observation indicates that the AN1 domain likely
predates the emergence of the A20 domain in the evolution of SAP proteins. The AN1 domain is more widely conserved among SAP proteins in plants, whereas the A20 domain is less prevalent as a standalone domain in SAP genes. However, it is important to note that this inference is based on the current knowledge and understanding of SAP genes in different plant species. Further research and analysis is necessary to gain a more comprehensive understanding of the evolutionary relationships and origins of the AN1 and A20 domains within the SAP gene family.

The variation in exon–intron numbers and conserved motifs within gene families often provides valuable insights into the evolutionary mechanisms of those gene families (*Abdullah-Zawawi et al., 2021*; *Song et al., 2022*). In the investigation of SAP genes across seven Compositae species, it was observed that members of the same subfamily exhibited similar exon–intron numbers and conserved motifs, suggesting shared biological functions within subfamilies. One characteristic of SAP family members is the tendency to exhibit a lack of introns in various plant species (*Vij & Tyagi, 2006*; *Xie et al., 2022*; *Jalal et al., 2022*). For example, approximately 61% of rice SAP genes have no introns, and 33% of soybean and rice SAP genes have only one intron (*Vij & Tyagi, 2006*). In *A. thaliana*, 64% of AtSAP genes have no introns, and 28% have one intron (*Vij & Tyagi, 2006*). The present study found that 72% of the SAP genes in Compositae species were intron-free, and 20% contained one intron. Previous studies in *A. thaliana* and rice have suggested that SAP genes without introns or with an intron deficiency are more likely to be involved in salt and drought stress responses (*Nguyen et al., 2016*; *Gimeno-Gilles et al., 2011*). The presence of the four SAP subfamily genes in multiple monocots and dicots suggests that the differentiation among SAP family members occurred prior to the divergence of these plant species. This indicates that the evolutionary history of the SAP gene family predates species differentiation. During biological evolution, gene duplication events and subsequent functional diversification have played significant roles in shaping genome and species evolution (*Bryant & Aves, 2011*). These mechanisms contribute to the expansion and diversification of gene families, allowing for the acquisition of new functions and adaptations over time. Sixteen pairs of tandem repeats were found in sunflower HaSAP genes, which may be the main reason for the expansion of the sunflower SAP family. This expansion is considered a survival strategy that enables sunflower to adapt to its environment. The HaSAP family of genes is relatively conserved evolutionarily, which helps maintain its functionality. Furthermore, multiple orthologous genes (more than 8 pairs) were identified between sunflower and other Compositae species (*A. lappa, C. endivia, C. intybus, C. morifolium, L. sativa var. angustata*, and *C. tinctorius*), indicating significant expansion of SAP genes during polyploidy events before these species differentiated. Overall, these findings provide valuable insights into the evolutionary mechanisms and expansion of the SAP gene family, highlighting the importance of gene duplication and diversification in shaping plant genomes and species adaptation.

Cis-regulatory elements are crucial molecular switches involved in regulating gene expression and controlling various biological processes, including responses to hormones, abiotic stresses, and developmental processes. These elements are typically found within the promoter regions of genes and control gene expression by interacting with

transcription factors and other regulatory proteins (*Mao et al., 2022*; *Mishra et al., 2018*; *Yamaguchi-Shinozaki & Shinozaki, 2005*). The present study found that the highest number of cis-acting elements in the HaSAP gene bind to ABA, a well-known regulator of abiotic stress responses in plants (*Asad et al., 2019*). The substantial changes in the transcript levels of selected HaSAP genes in response to drought and salt stress observed in the qRT-PCR analysis provide strong evidence for the potential involvement of these genes in plant drought and salt stress responses. The changes in gene expression observed suggest that the HaSAP genes may play a role in the molecular response of sunflower plants to these abiotic stresses. Previous studies have shown that the overexpression of *AtSAP10* and *AtSAP13* in *A. thaliana* enhances the plant's tolerance to various toxic metals (*Dixit & Dhankher, 2011*; *Dixit et al., 2018*). Similarly, the overexpression of soybean *SAP16* in both *A. thaliana* and soybean plants has been shown to increase the tolerance of these plants to drought and salt stress (*Zhang et al., 2019*). In addition, the overexpression of the *M. truncatula SAP1* gene increased transgenic tobacco's ability to resist abiotic stresses (*Charrier et al., 2013*) and the overexpression of *MdSAP15* enhanced the osmosis and drought resistance of transgenic *A. thaliana* plants compared to wild-type plants (*Dong et al., 2018*). In sunflower plants, the expression profiles of *HaSAP1, HaSAP21*, and *HaSAP23* remained consistent under salt and drought stress conditions, indicating that these genes may play a role in the salt tolerance and drought resistance traits of sunflower plants. The altered expression of HaSAP genes in response to drought and salt stress may trigger the expression of ABA-related genes, leading to improved resistance to stress in sunflower plants. Overall, these findings highlight the potential involvement of SAP genes in the abiotic stress responses of plants and suggest that the regulation of gene expression through cis-regulatory elements and ABA-related pathways plays a role in the physiological and developmental responses of plants to environmental challenges.

## CONCLUSIONS

This study comprehensively analyzed the number, structure, collinearity, and phylogeny of the SAP gene family in 15 plant species. The phylogenetic analysis revealed that the genes in the SAP family could be divided into three subgroups, and these subgroup genes were found in both monocots and dicots, suggesting that the differentiation of SAP family members occurred earlier than species differentiation. Notably, the sunflower and *Z. mays* SAP genes exhibited the highest number of collinear gene pairs, indicating a close relationship between these two species within the Compositae family. The promoter analysis indicated the presence of several cis-acting elements related to ABA, suggesting that SAP genes may be involved in ABA-mediated stress responses in sunflower plants. The expression analysis revealed that *HaSAP1, HaSAP21*, and *HaSAP23* displayed consistent expression patterns under salt and drought stress conditions. This consistency suggests that these genes may play a role in the salt tolerance and drought resistance of sunflower plants. Overall, the results of this study provide important insights and clues for further research on the role of SAP genes in the salt tolerance and drought resistance of sunflower plants. These findings help elucidate the molecular mechanisms underlying the stress responses of sunflower plants and can guide future investigations in this area.

### Funding

This work was supported by the Natural Science Foundation of Inner Mongolia Autonomous Region (2023LHMS03033), the Sunflower Industry Technology Innovation and Promotion System of Inner Mongolia Autonomous Region (CARS-IMAR-4) and the Science and Technology Research Project of Hetao College (HYHB202302). The funders had no role in study design, data collection and analysis, decision to publish, or preparation of the manuscript.

### Grant Disclosures

The following grant information was disclosed by the authors:
Natural Science Foundation of Inner Mongolia Autonomous Region: 2023LHMS03033.
Sunflower Industry Technology Innovation and Promotion System of Inner Mongolia Autonomous Region: ARS-IMAR-4.
Science and Technology Research Project of Hetao College: HYHB202302.

### Competing Interests

The authors declare that they have no competing interests.

### Author Contributions

- Chun Zhang conceived and designed the experiments, performed the experiments, analyzed the data, prepared figures and/or tables, authored or reviewed drafts of the article, and approved the final draft.
- Xiaohong Zhang performed the experiments, prepared figures and/or tables, and approved the final draft.
- Yue Wu performed the experiments, prepared figures and/or tables, and approved the final draft.
- Xiang Li performed the experiments, prepared figures and/or tables, and approved the final draft.
- Chao Du analyzed the data, prepared figures and/or tables, and approved the final draft.
- Na Di analyzed the data, prepared figures and/or tables, and approved the final draft.
- Yang Chen conceived and designed the experiments, authored or reviewed drafts of the article, and approved the final draft.

### DNA Deposition

The following information was supplied regarding the deposition of DNA sequences:
The RNA-seq data presented in the study are available at NCBI: PRJNA866668.

### Data Availability

The raw data is available in the Supplemental Files.

## Supplemental Information

Supplemental information for this article can be found online at http://dx.doi.org/10.7717/peerj.17808#supplemental-information.

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
