# Peer review of "Genome-wide identification and evolution of the SAP gene family in sunflower (Helianthus annuus L.) and expression analysis under salt and drought stress"

_PeerJ, doi:10.7717/peerj.17808_

## Round 0.1 · original submission · Major Revisions

We have received three different reviews on this work. There are list of comments demanding the text update. Please check annotated file kindly provided by reviewer #3.

**Language Note:** The review process has identified that the English language must be improved. PeerJ can provide language editing services - please contact us at [email protected] for pricing (be sure to provide your manuscript number and title). Alternatively, you should make your own arrangements to improve the language quality and provide details in your response letter. – PeerJ Staff

Reviewer 1 ·

Basic reporting

The SAP gene family plays a pivotal role in plants' tolerance to adverse environmental conditions. This research specifically focuses on the SAP gene family in sunflowers, providing valuable insights for enhancing the stress resistance of sunflower. In this study, 27 SAP genes were successfully identified from sunflowers, and their gene structures, conserved protein motifs, evolutionary relationships, and expression patterns were comprehensively analyzed. The findings in this study lay a solid foundation for further studies on the functions of the SAP genes in sunflower. Nevertheless, there are still many flaws in the manuscript that need to be revised and improved.

Experimental design

no comment

Validity of the findings

no comment

Additional comments

Some shortcomings in the manuscript are as follows:

(1) In the results section, the result analysis are not sufficiently in-depth and merely consist of descriptive language. In addition, the readability of the results section in the manuscript needs to be further improved.

(2) In Line162-172,the sunflower SAP genes are divided into four groups based on the phylogenetic tree. However, such evolutionary grouping is questionable. As is commonly known, genes within the same group should have a closer evolutionary relationship than those between different groups. However, as can be seen in Figure 1, some SAP genes in Group 3 are clustered within the same branch as SAP genes in Group 4, while being distant from other SAP genes in Group 3. Obviously, this is not in line with common understanding. Therefore, it is recommended to regroup the SAP genes based on the evolutionary tree.

(3)In Line 181-184, “and the most collinearity gene pairs between sunflower SAP and Z. mays are closer to each other in the branches of the phylogenetic tree, and Z. mays contains only 9 SAP genes, which indicates that sunflower and Z. mays the SAP gene in may be more functionally similar.” This sentence tells us that the synteny between sunflower and maize is better than that with other plants. This seems questionable. As we know, nine species were involved in the synteny analysis (sunflower, licorice, rice, maize, Arabidopsis, canola, watermelon, tomato, and grape). Among them, rice and maize are monocots, while the other plants are dicots. It is somewhat surprising that sunflower, as a dicot, exhibits better synteny with maize, a monocot. Please confirm this again.

(4) In qRT-PCR analysis, using the 0h time point as the control is not entirely reasonable. Even without salt or drought stresses, there can be differences in the expression of SAP genes at 6h or 12h compared to the 0h time point. Therefore, it is difficult to ascertain whether the differences in gene expression are solely induced by salt stress or drought stress.

(5) In this manuscript, especially in the results section, the language needs further revision.

(6) In the manuscript, some of the figures are not clear enough. Please redraw them.

Reviewer 2 ·

Basic reporting

The present study comprehensively investigates the identification, evolution, and expression analysis of members within the sunflower SAP family, with the aim of elucidating their evolutionary history and potential functions. The methodology employed is appropriate, research content comprehensive, and figure and table representations sound. I recommend that the authors further addressed or revised the following issues prior to publication.
1. Abstract
The abstract is excessively verbose. It is unnecessary to enumerate the specific names of the Compositae and other species; instead, focus on presenting the main results. The final sentence "in conclusion" seems more like an indication of the study's significance rather than a true conclusion.
2. Introduction
1) Line 44-47: These two sentences should be moved to the first paragraph. Additionally, it may be beneficial to follow up with a statement regarding the research progress of this family in evolution and provide an explanation for its evolutionary origin.
2) Line 47-58: This sentence does not establish a logical relationship with the preceding sentences, and fails to indicate functional importance adequately.
3) The third paragraph should discuss sunflower research in relation to salt and drought stress in order to align with the aim of study.
3. M&M
1) I suggest placing “Plant material” section before RT-qPCR, as it is less closely related to other sections.
2) Line 99-100: Latin names should be italicized.
3) Essential parameters for reconstructing the phylogenetic tree were required.
4) Line 141: Actin should be italicized.
5) Please list the statistical method used for analyzing differences' significance.

4. Results
1) In my opinion, there is no need for separating parts two and three; instead, integrate both evolutionary analyses into one section, while keeping gene structure and motif analysis separate as its own part.
2) Why are there no error bars present in Figure 8-10's leftmost column? Also, do asterisks represent significance compared with control (leftmost column)? Please consider using different letters instead of asterisks when representing significant differences among samples.

Experimental design

no comment

Validity of the findings

no comment

Additional comments

no comment

Reviewer 3 ·

Basic reporting

no comment

Experimental design

no comment

Validity of the findings

no comment

Additional comments

no comment

Annotated reviews are not available for download in order to protect the identity of reviewers who chose to remain anonymous.

---

## Round 0.2 · Minor Revisions

Thanks for the update and manuscript revision. This manuscript version got positive comments. However, please check the very minor technical remarks by reviewer #3. Please see the annotated file attached.

Reviewer 1 ·

Basic reporting

The authors carefully revised the manuscript in accordance with the review comments and, in addition, responded to each question from the review comments.

Experimental design

no comment

Validity of the findings

no comment

Additional comments

no comment

Reviewer 2 ·

Basic reporting

The authors have well addressed all my issues.

Experimental design

no comment

Validity of the findings

no comment

Additional comments

no comment

Reviewer 3 ·

Basic reporting

The author has made careful revisions and answers to most of the previous questions, and I fully agree with the results of the author's revisions.
But the question “In the results part, the transcriptome only had salt stress data, but no drought stress and tissue-specific expression data, and salt stress data were only available at three time points. In addition, are the materials and processing methods of the transcriptome samples the same as the qRT-PCR samples in this experiment? Why did the qRT-PCR experiment in this study screen 10 genes obtained using transcriptome data, while the other genes were not further analyzed? is not still clearly answered, I hope the author can further explain.
In addition, the questions “Line245. "(Fig. 7)" should be amended as "(Figure 7)" and be consistent with the rest of the manuscript.” After the modification, how did it become Figure 6? Please check it again. And should the note in line 341 not be deleted?
My decision: Once the authors explains my concerns and incorporate the changes in their manuscript, I would be happy to recommend it for its acceptance.

Experimental design

no

Validity of the findings

no

Additional comments

no

Annotated reviews are not available for download in order to protect the identity of reviewers who chose to remain anonymous.

---

## Round 0.3 · accepted · Accept

Thanks for the manuscript update and reply. There are no more critical remarks from the reviewers.